

# The gut microbiota in the common kestrel (*Falco tinnunculus*): a report from the Beijing Raptor Rescue Center

Yu Guan, Hongfang Wang, Yinan Gong, Jianping Ge and Lei Bao

Ministry of Education Key Laboratory for Biodiversity Science and Ecological Engineering and College of Life Science, Beijing Normal University, Beijing, China

## ABSTRACT

As a complex microecological system, the gut microbiota plays crucial roles in many aspects, including immunology, physiology and development. The specific function and mechanism of the gut microbiota in birds are distinct due to their body structure, physiological attributes and life history. Data on the gut microbiota of the common kestrel, a second-class protected animal species in China, are currently scarce. With high-throughput sequencing technology, we characterized the bacterial community of the gut from nine fecal samples from a wounded common kestrel by sequencing the V3-V4 region of the 16S ribosomal RNA gene. Our results showed that *Proteobacteria* (41.078%), *Firmicutes* (40.923%) and *Actinobacteria* (11.191%) were the most predominant phyla. *Lactobacillus* (20.563%) was the most dominant genus, followed by *Escherichia-Shigella* (17.588%) and *Acinetobacter* (5.956%). Our results would offer fundamental data and direction for the wildlife rescue.

## INTRODUCTION

Recent research on host-associated gut microbial communities have revealed their important roles in immunology, physiology and development (*Guarner & Malagelada, 2003*; *Nicholson, Holmes & Wilson, 2005*), as well as several basic and critical processes, such as nutrient absorption and vitamins synthesis in both human and animals (*Fukuda & Ohno, 2014*; *Kau et al., 2011*; *Omahony et al., 2015*). Gut microbiota analysis of wild animals is becoming a new method that may provide information for wildlife rescue and animal husbandry. Reports concerning the gut microbiota of other avian species, such as Cooper's hawk (*Accipiter cooperii*) (*Taylor et al., 2019*), bar-headed geese (*Anser indicus*) (*Wang et al., 2017*), hooded crane (*Grus monacha*) (*Zhao et al., 2017*), Western Gull (*Larus occidentalis*) (*Cockerham et al., 2019*), herring gull (*Larus argentatus*) (*Fuirst et al., 2018*) and black-legged kittiwake (*Rissa tridactyla*) (*Van Dongen et al., 2013*), have increased rapidly. The specific function and mechanism of the gut microbiota in birds are distinct due to their body structure, physiological attributes and life history (*Kobayashi, 1969*; *Williams & Tieleman, 2005*; *Winter, Johnson & Shaffer, 2006*). For example, for most birds, a stable body temperature above ambient temperature ensures a high metabolic

Corresponding author
Lei Bao, baolei@bnu.edu.cn

rate for the birds needed for flight (*O'Mara et al., 2017*; *Schleucher, 2002*; *Smit et al., 2016*). Streamlined bodies, efficient breathing patterns and relatively short gastrointestinal tracts are also special attributes (*Klasing, 1999*; *Orosz & Lichtenberger, 2011*). Meanwhile, the birds' ability to fly sets them apart from other animals, altering their intestinal microbiota to some extent. However, as a research focus, data on the gut microbiota of the common kestrel are currently very scarce.

The common kestrel (*Falco tinnunculus*) is a small raptor that belongs to *Falconidae*, which is a family of diurnal birds of prey, including falcons and kestrels. A total of 12 subspecies for common kestrel are distributed widely from the Palearctic to Oriental regions (*Cramp & Brooks, 1992*). Although listed in the least concern (LC) class by the International Union for Conservation of Nature (IUCN) (*BirdLife International, 2016*), the common kestrel was listed as state second-class protected animals (Defined by the LAW OF THE PEOPLE'S REPUBLIC OF CHINA ON THE PROTECTION OF WILDLIFE, Chapter II, Article 9) in China. The common kestrel is a typical opportunistic forager that catches small and medium-sized animals, including small mammals, birds, reptiles and some invertebrates (*Anthony, 1993*; *Aparicio, 2000*; *Village, 2010*). Insects such as grasshoppers and dragonflies were also identified in the diet of the common kestrel (*Geng et al., 2009*). As generalist predators, common kestrels choose distinct predatory strategies when non-breeding and breeding to minimize the expenditure of energy, such as the strategy of the low-cost low-profit technique of perch-hunting in winter, while maximized daily energy gain in summer (*Costantini et al., 2005*; *Masman, Daan & Dijkstra, 1988*).

Previous studies on common kestrels were comprehensive, such as those on diet and prey selection (*Geng et al., 2009*; *Kirkwood, 1980*; *Korpimäki, 1985*; *Lihu et al., 2007*; *Souttou et al., 2007*; *Van Zyl, 1994*), behavior and diseases (*Aschwanden, Birrer & Jenni, 2005*; *Bustamante, 1994*; *Hille, Nash & Krone, 2007*), and genetic variation and diversity (*Nesje et al., 2000*; *Padilla et al., 2009*; *Riegert, Fainová & Bystřická, 2010*; *Zhang, Liu & Song, 2008*). As common raptors around the whole world, as well as the important predators in food chains, common kestrels should be studied more deeply with the newer methods and techniques.

The aim of this study was to characterize the bacterial community of the gut by sequencing the V3-V4 region of the 16S rRNA gene of a wounded common kestrel. The data we obtained could provide basic information for further conservation and rescue of wild common kestrels.

## MATERIALS & METHODS

### Fecal samples collection

This study is of a single kestrel sampled multiple times with feces at Beijing Raptor Rescue Center (BRRC). The injured common kestrel that could not fly was found in the Fengtai district by a rescuer on June 22nd, 2019 and then taken to the BRRC for professional rescue. The wounded common kestrel was carefully treated with several surgeries and drug therapies. Nine fecal samples (E1–E9) that may reflect the actual state of its health were collected from the common kestrel after relevant treatments on different days. The samples
collection information and medical records of the common kestrel were shown in Tables S1–S3 respectively. All samples were transported immediately into the laboratory in an ice box and ultimately stored at −80 °C for further bacterial studies.

## DNA extraction and PCR amplification

Microbial DNA was extracted from fresh fecal samples using an E.Z.N.A.® Stool DNA Kit (Omega Bio-tek, Norcross, GA, U.S.) according to the manufacturer's protocols. The V3–V4 region of the bacterial 16S ribosomal RNA gene was amplified by PCR (95 °C for 3 min; followed by 25 cycles at 95 °C for 30 s, 55 °C for 30 s, and 72 °C for 30 s; and a final extension at 72 °C for 5 min) using the primers 338F (5′-barcode-ACTCCTACGGGAGGCAGCAG-3′) and 806R (5′-GGACTACHVGGGTWTCTAAT-3′), where the barcode is an eight-base sequence unique to each sample. PCRs were performed in triplicate in a 20 μL mixture containing 4 μL of 5× FastPfu Buffer, 2 μL of 2.5 mM dNTPs, 0.8 μL of each primer (5 μM), 0.4 μL of FastPfu Polymerase, and 10 ng of template DNA.

## Illumina MiSeq sequencing

Amplicons were extracted from 2% agarose gels and purified using an AxyPrep DNA Gel Extraction Kit (Axygen Biosciences, Union City, CA, U.S.) according to the manufacturer's instructions and quantified using QuantiFluor™ -ST (Promega, U.S.). Purified amplicons were pooled in equimolar amounts and paired-end sequenced (2× 250) on an Illumina MiSeq platform according to standard protocols.

## Processing of sequencing data

Raw fastq files were demultiplexed and quality-filtered using QIIME (version 1.17) (*Caporaso et al., 2010*) with the following criteria. (i) The 300 bp reads were truncated at any site receiving an average quality score <20 over a 50 bp sliding window, discarding the truncated reads that were shorter than 50 bp. (ii) Exact barcode matching, 2 nucleotide mismatches in primer matching, and reads containing ambiguous characters were removed. (iii) Only sequences that overlapped longer than 10 bp were assembled according to their overlap sequence. Reads that could not be assembled were discarded.

Operational taxonomic units (OTUs) were clustered with a 97% similarity cutoff using UPARSE (version 7.1 http://drive5.com/uparse/), and chimeric sequences were identified and removed using UCHIME (*Edgar et al., 2011*). The taxonomy of each 16S rRNA gene sequence was analyzed by RDP Classifier (http://rdp.cme.msu.edu/) against the SILVA (SSU132)16S rRNA database using a confidence threshold of 70% (*Amato et al., 2013*).

## Data analysis

All the indices of alpha diversity, including Chao, ACE, Shannon, Simpson, and coverage, and the analysis of beta diversity were calculated with QIIME. The rarefaction curves, rank abundance curves, and stacked histogram of relative abundance were displayed with R (*R Core Team, 2015*).

The hierarchical clustering trees were built using UPGMA (unweighted pair-group method with arithmetic mean) based on weighted and unweighted distance matrices at different levels. Principal coordinate analysis (PCoA) was calculated and displayed using QIIME and R, as well as hierarchical clustering trees.

This study was performed in accordance with the recommendations of the Animal Ethics Review Committee of Beijing Normal University (approval reference number: CLS-EAW-2019-026).

## RESULTS

### Overall sequencing data

A total of 28 phyla, 70 classes, 183 orders, 329 families and 681 genera were detected among the gastrointestinal bacterial communities. There were altogether 389,474 reads obtained and classified into 1673 OTUs at the 0.97 sequence identity cut-off in 9 fecal samples from a common kestrel.

Alpha diversity indices (including Sobs, Shannon, Simpson, ACE, Chao and coverage) of each sample are shown in Table 1. The Sobs and Shannon index of all samples are shown in Fig. 1. Additionally, the rarefaction curves (A) and the rank abundance curves (B) are shown in Fig. S1, which indicated that the number of OTUs for further analysis was reasonable, as well as the abundance of species in common kestrel feces. The total sequences, total bases and OTU distributions of all samples are shown in Tables S4 and S5.

### Bacterial composition and relative abundance

At the phylum level of the gut microbiota in the common kestrel, the most predominant phylum was *Proteobacteria* (41.078%), followed by *Firmicutes* (40.923%), *Actinobacteria* (11.191%) and *Bacteroidetes* (3.821%). In addition to *Tenericutes* (0.178%) and *Verrucomicrobia* (0.162%), *Patescibacteria* (0.543%) and *Deinococcus-Thermus* (0.504%) were also ranked in the top 10 species in the common kestrel fecal microbiota (Table 2).

The top five families in the gut microbiota were *Lactobacillaceae* (20.563%), *Enterobacteriaceae* (18.346%), *Moraxellaceae* (6.733%), *Bifidobacteriaceae* (5.624%) and *Burkholderiaceae* (4.752%).

At the genus level, *Lactobacillus* (20.563%), *Escherichia-Shigella* (17.588%) and *Acinetobacter* (5.956%) were the most dominant genera. These were followed by *Bifidobacterium* (5.624%) and *Enterococcus* (4.024%) (Table 3). These five genera in the total gut microbiota of several samples accounted for a small proportion, such as for E5 (28.755%) and E6 (10.905%) and especially for E4 (2.861%), while the largest proportion was 98.416% in E1.

The stacked histogram of relative abundance for species is also demonstrated in Fig. 2 at the phylum (A) and genus (B) levels, which could intuitively represent the basic bacterial composition and relative abundance. The community structures of E1 and E9 were more similar than those of the other feces samples at both levels.

The hierarchical clustering trees showed the similarity of community structure among different samples, which were generated by UPGMA (unweighted pair-group method with arithmetic mean) with the unweighted UniFrac (Fig. 3A) and weighted UniFrac (Fig. 3B) distance matrixes. Although the fecal samples were collected from the common kestrel in chronological order (E1–E9) of therapy treatments, no distinct or obvious clustering relationships are discernable in Fig. 3.

**Table 1  Alpha diversity of gut microbiota in Common Kestrel feces.**

| Sample | Sobs | Shannon | Simpson | Ace | Chao | Coverage |
|--------|------|---------|---------|---------|---------|----------|
| E1 | 66 | 0.596 | 0.788 | 78.114 | 73.583 | 1.000 |
| E2 | 649 | 2.780 | 0.204 | 674.412 | 672.193 | 0.998 |
| E3 | 515 | 2.965 | 0.184 | 524.452 | 522.519 | 0.999 |
| E4 | 578 | 4.233 | 0.053 | 594.498 | 594.050 | 0.999 |
| E5 | 235 | 3.285 | 0.057 | 448.368 | 378.103 | 0.997 |
| E6 | 476 | 4.802 | 0.020 | 479.110 | 480.091 | 1.000 |
| E7 | 263 | 1.604 | 0.399 | 292.553 | 281.800 | 0.999 |
| E8 | 317 | 2.706 | 0.143 | 364.651 | 359.519 | 0.998 |
| E9 | 317 | 2.374 | 0.335 | 330.906 | 331.607 | 0.999 |

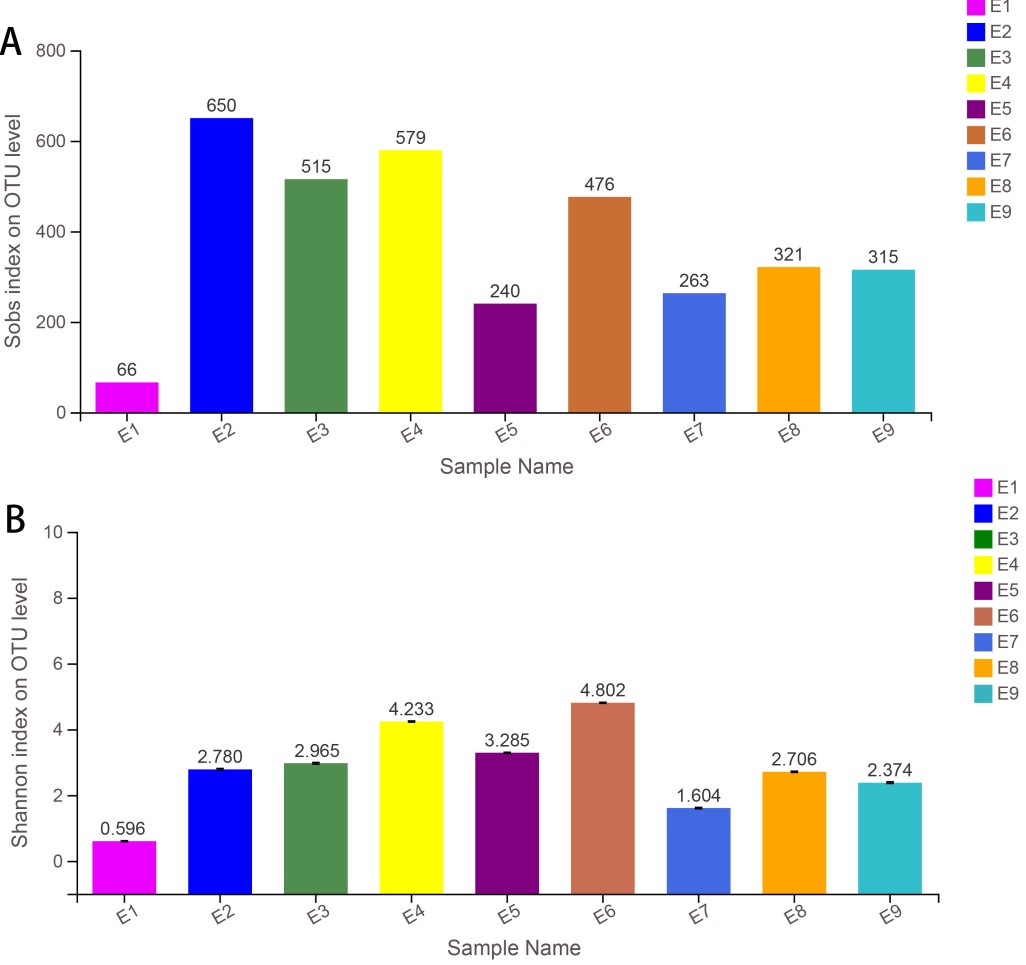

**Figure 1  Sobs index (A) and the Shannon index (B) of samples.**

**Table 2** **The relative abundance of species in gut microbiota of Common Kestrel at phylum level.** The names of phyla in Table 2 represented *Proteobacteria, Firmicutes, Actinobacteria, Bacteroidetes, Patescibacteria, Deinococcus-Thermus, unclassified_K_norank_d_Bacteria, Tenericutes, Verrucomicrobia*, respectively.

| Sample | Pro | Fir | Act | Bac | Pat | Dei | unc | Ten | Ver | Others |
|---|---|---|---|---|---|---|---|---|---|---|
| E1 | 88.630% | 10.634% | 0.623% | 0.006% | 0.003% | 0.000% | 0.096% | 0.000% | 0.006% | 0.003% |
| E2 | 13.211% | 79.816% | 2.376% | 1.085% | 0.085% | 0.065% | 0.361% | 1.291% | 0.017% | 1.694% |
| E3 | 50.540% | 39.567% | 3.286% | 1.502% | 1.857% | 0.087% | 1.553% | 0.121% | 0.011% | 1.474% |
| E4 | 73.770% | 3.574% | 8.602% | 5.950% | 1.719% | 3.960% | 0.220% | 0.008% | 0.158% | 2.038% |
| E5 | 27.797% | 43.152% | 10.694% | 18.166% | 0.104% | 0.042% | 0.006% | 0.000% | 0.000% | 0.039% |
| E6 | 36.410% | 30.610% | 20.330% | 5.572% | 0.944% | 0.324% | 1.511% | 0.135% | 1.223% | 2.940% |
| E7 | 5.000% | 73.097% | 20.770% | 0.676% | 0.003% | 0.000% | 0.076% | 0.000% | 0.006% | 0.372% |
| E8 | 11.832% | 59.652% | 27.752% | 0.369% | 0.073% | 0.003% | 0.031% | 0.000% | 0.031% | 0.256% |
| E9 | 62.507% | 28.205% | 6.285% | 1.065% | 0.096% | 0.056% | 0.671% | 0.045% | 0.006% | 1.063% |
| Mean | 41.078% | 40.923% | 11.191% | 3.821% | 0.543% | 0.504% | 0.503% | 0.178% | 0.162% | 1.098% |

**Table 3** **The relative abundance of species in gut microbiota of Common Kestrel at genus level.** The names of phyla in Table 3 represented *Lactobacillus, Escherichia-Shigella, Acinetobacter, Bifidobacterium, Enterococcus, Clostridium_sensu_stricto_1, Paracoccus, Burkholderia- Caballeronia-Paraburkholderia, Glutamicibacter*, respectively.

| Sample | Lac | Esc | Aci | Bif | Ent | Clo | Par | Bur | Glu | Others |
|---|---|---|---|---|---|---|---|---|---|---|
| E1 | 6.618% | 88.610% | 0.011% | 0.037% | 3.140% | 0.023% | 0.000% | 0.000% | 0.000% | 1.561% |
| E2 | 68.336% | 1.787% | 0.581% | 0.034% | 2.528% | 0.042% | 0.017% | 5.567% | 0.042% | 21.066% |
| E3 | 24.037% | 0.862% | 38.448% | 0.093% | 0.448% | 0.707% | 0.101% | 5.544% | 0.023% | 29.736% |
| E4 | 0.392% | 0.166% | 2.153% | 0.031% | 0.118% | 0.214% | 19.488% | 0.854% | 0.860% | 75.724% |
| E5 | 0.693% | 7.962% | 5.040% | 0.034% | 15.026% | 0.011% | 0.149% | 0.014% | 9.411% | 61.659% |
| E6 | 1.356% | 0.130% | 5.823% | 3.086% | 0.510% | 1.043% | 6.023% | 4.445% | 6.113% | 71.470% |
| E7 | 66.056% | 0.054% | 0.211% | 20.356% | 2.120% | 0.192% | 0.031% | 2.841% | 0.011% | 8.129% |
| E8 | 9.589% | 1.536% | 0.536% | 25.502% | 8.988% | 28.177% | 0.054% | 2.060% | 0.099% | 23.459% |
| E9 | 7.988% | 57.183% | 0.798% | 1.446% | 3.340% | 1.866% | 0.273% | 0.347% | 0.000% | 26.759% |
| Mean | 20.563% | 17.588% | 5.956% | 5.624% | 4.024% | 3.586% | 2.904% | 2.408% | 1.840% | 35.507% |

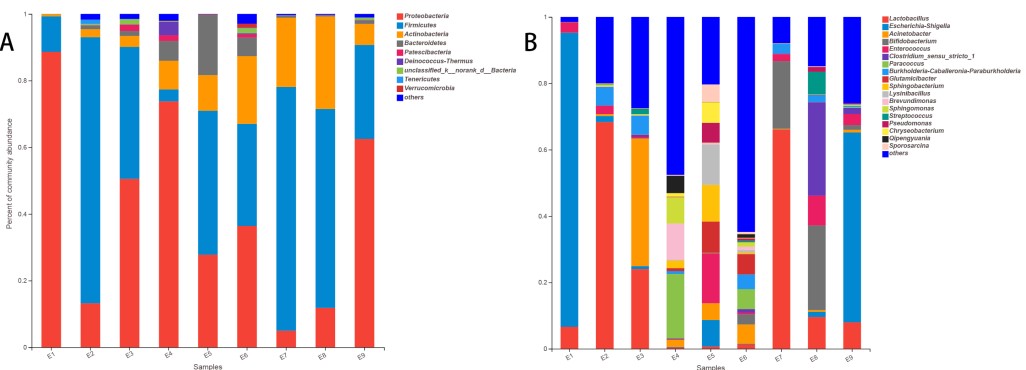

**Figure 2** **The histogram of relative abundance for species in Common Kestrel at phylum (A) and genus (B) level.**

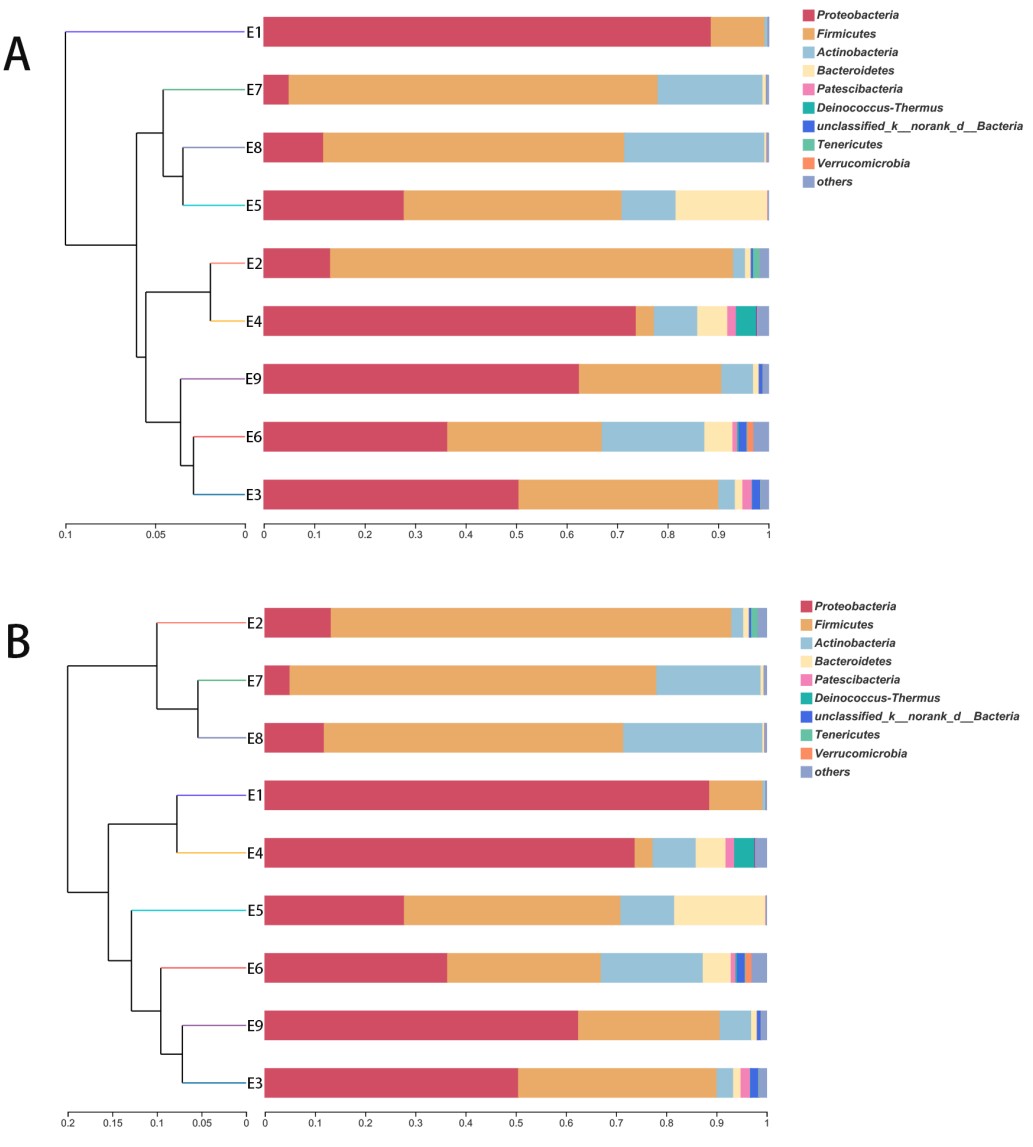

**Figure 3** **The hierarchical clustering trees.** (A) and (B) were generated based on unweighted and weighted distance matrix at phylum level, respectively.

## Discrepancy of community composition

To further demonstrate the differences in community composition among the nine samples, principal coordinates analysis (PCoA) was applied (Fig. 4). For PCoA, we chose the same two distance matrices (unweighted UniFrac in Fig. 4A and weighted UniFrac in Fig. 4B) as above to analyze the discrepancies. The results in Fig. 4 were similar to those in Fig. 3, in which all samples scattered dispersedly, suggesting that variation in the composition of the gut microbiota of the common kestrel was not obvious over time.

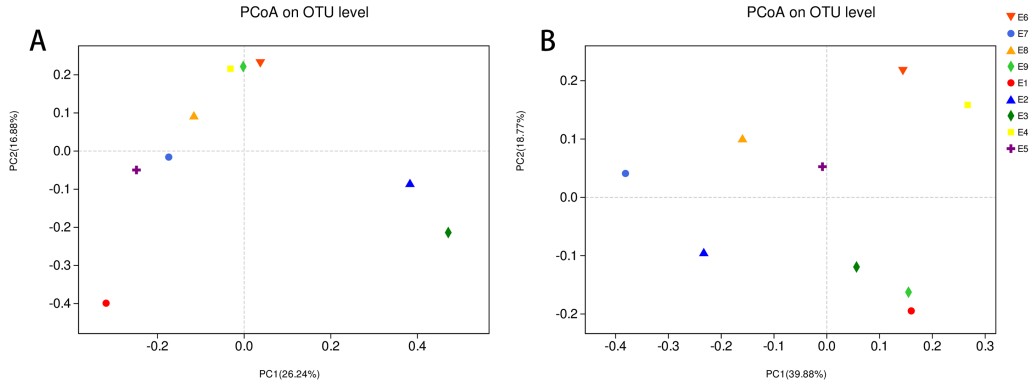

**Figure 4** **PCoA of the bacterial population structures.** The different shape with colors represented all samples of Common Kestrel respectively. For PCoA, (A) was generated with unweighted Unifrac distance while (B) used weighted Unifrac distance.

# DISCUSSION

Knowledge and comprehension concerning gut microbiota have continued to progressively develop with relevant techniques over the past decade (*Guarner, 2014*; *Li et al., 2014*; *Qin et al., 2010*). The application of analysis for intestinal microecology continues to be also a research focus in the field of wildlife rescue.

The common kestrel (*Falco tinnunculus*) is listed as a second-class protected animal species in China. Although research concerning avian species, including the common kestrel, has been increasing gradually, the available data on the gut microbiota in the common kestrel were currently unknown.

We characterized the basic composition and structure of the gut microbiota from a wounded common kestrel in this study, which was rescued by the Beijing Raptor Rescue Center (BRRC).

In general, the overall community structure of the gut microbiota in this common kestrel was in accordance with previous relevant characterizations in birds, such as Cooper's hawks (*Taylor et al., 2019*), bar-headed geese (*Wang et al., 2017*), hooded cranes (*Zhao et al., 2017*) and swan geese (*Wang et al., 2016*), which included *Proteobacteria*, *Firmicutes*, *Actinobacteria* and *Bacteroidetes*.

The most predominant phylum in the fecal gut microbiota of the common kestrel was *Proteobacteria* (41.078%), which ranked after *Firmicutes* in other birds, such as cockatiels (*Nymphicus hollandicus*) (*Alcaraz, Hernández & Peimbert, 2016*) and black-legged kittiwakes (*Van Dongen et al., 2013*). This crucial phylum plays many valuable roles. For instance, *Proteobacteria* is beneficial for the giant panda, which can degrade lignin in its major food resource (*Fang et al., 2012*). Additionally, it has been reported that *Proteobacteria* is also the most dominant phylum in obese dogs (*Park et al., 2015*). The specific function of this phylum could be distinct in birds due to their unique physiological traits, as well as their developmental strategies (*Kohl, 2012*). However, the high relative abundance of *Proteobacteria* in the total bacterial community was observed mainly in

several samples that were collected during surgeries or drug treatments, such as E1 and E4. Sample E1 was collected on 23rd June that the day after the kestrel rescued from the wild. On 22nd June, the kestrel was bandaged with silver sulfadiazine cream (SSD), also subcutaneously injected with 10 ml and orally administered with 4ml lactated ringer's solution (LRS) respectively. The increased level of *Proteobacteria* was associated with some cardiovascular events, inflammation and inflammatory bowel disease (*Amar et al., 2013*; *Carvalho et al., 2012*). Although the kestrel's weight increased 34 grams when E4 was collected, it just ate a mouse's head. Combined with the status when the kestrel was rescued, we speculated that the increased proportion of *Proteobacteria* may reflect its food consumption or gastrointestinal status to some extent. Environmental influential factors, as well as dietary changes, should also be considered an important index that could result in variations in the relative abundance of species in the gut microbiota (*De Filippo et al., 2010*; *Scott et al., 2013*).

Furthermore, the dominant genera within *Proteobacteria* in our study were *Escherichia-Shigella* (17.588%), *Acinetobacter* (5.956%), *Paracoccus* (2.904%) and *Burkholderia-Caballeronia-Paraburkholderia* (2.408%). *Escherichia-Shigella* is a common pathogenic bacterium that can cause diarrhea in humans (*Hermes et al., 2009*). The main cause for the high relative abundance of *Escherichia-Shigella* was the E1 (88.610%) sample, which suggested indirectly that the physical condition of the common kestrel was not normal when it was rescued by staff from the BRRC. This result was also consistent with the actual state of this wounded common kestrel that we observed (Table S3).

Although *Firmicutes* (40.923%) ranked after *Proteobacteria*, its actual relative abundance was only slightly lower than that in the common kestrel. As a common phylum of the gut microbiota, *Firmicutes* exists widely in both mammals and birds, and this ancient symbiosis may be linked to the common ancestor of amniotes (*Costello et al., 2010*; *Kohl, 2012*). *Firmicutes* can provide certain energy for the host through catabolizing complex carbohydrates, sugar, and even by digesting fiber in some species (*Costa et al., 2012*; *Flint et al., 2008*; *Guan et al., 2017*).

The dominant genera in *Firmicutes* were *Lactobacillus* (20.563%), *Enterococcus* (4.024%) and *Clostridium_sensu_stricto_1* (3.586%). The relative abundance of *Enterococcus* in E5 (15.026%) contributed to the highest ranking of this genus. *Enterococcus* is not regarded as a pathogenic bacterium due to its harmlessness and can even be used as a normal food additive in related industries (*Fisher & Phillips, 2009*; *Moreno et al., 2006*). *Enterococcus* species are also considered common nosocomial pathogens that can cause a high death rate (*Lopes et al., 2005*). Meanwhile, these species are also associated with certain infections, including neonatal infections, intraabdominal and pelvic infections, as well as the nosocomial infections and superinfections (*Murray, 1990*). Coincidentally, prior to the collection of sample E5, the kestrel was anesthetized for the treatment of the right tarsometatarsus injury. The right digit tendon of the kestrel was exposed before managing the wound, without any function. Although ensuring the sterile conditions, we inferred that the kestrel was infected by certain bacteria during the surgery. The BRRC could be regarded as a specific hospital for raptor, which could explain the high proportion of *Enterococcus* in the fecal samples of this common kestrel. However, this genus should be given sufficient attention

in subsequent studies with additional samples from different individuals. The abundance of *Clostridium* increases as more protein is digested (*Lubbs et al., 2009*). *Clostridium difficile* has been reported to be associated with certain diseases, such as diarrhea and severely life-threatening pseudomembranous colitis (*Kuijper, Coignard & Tull, 2006*; *Pepin et al., 2004*). The high relative abundance of this genus also resulted primarily from certain samples (E8, 28.177%), similar to the *Enterococcus* mentioned above. And it's remarkable that the collection of sample E8 was in the same situation as E5. On 13th July, the kestrel also underwent surgery under anesthesia. While E5 was collected, the kestrel's status was still normal according to relevant records. These results indicated that the high relative abundance of certain pathogens may not show any symptoms of illness for the kestrel. In general, the abnormal situation of E5 and E8 still need to be paid enough attention. Moreover, to minimize the influences due to the individual differences, more samples from different individuals should be collected for further study.

The third dominant phylum in the gut microbiota in our study was *Actinobacteria* (11.191%), which was also detected in other species, such as turkeys (*Meleagris gallopavo*) (*Wilkinson et al., 2017*) and Leach's storm petrel (*Oceanodroma leucorhoa*) (*Pearce et al., 2017*). The relative abundance of *Actinobacteria* varied in different species, such as house cats (7.30%) and dogs (1.8%) (*Handl et al., 2011*), but only accounted for 0.53% in wolves (*Wu et al., 2017*). Within this phylum, *Bifidobacterium* (5.624%) and *Glutamicibacter* (1.840%) were the primary genera. The presence of *Bifidobacterium* is closely related to the utilization of glycans produced by the host, as well as oligosaccharides in human milk (*Sela et al., 2008*; *Turroni et al., 2010*). Noticeably, *Bifidobacterium thermophilum* was reported to be used through oral administration for chickens to resist *E. coli* infection (*Kobayashi et al., 2002*). The detection and application of *Bifidobacterium*, especially for the rescue of many rare avian species, would be worth considering for curing various diseases in the future.

Additionally, the relative abundance of *Bacteroidetes* was 3.821% in this study, which consisted mainly of *Sphingobacterium*. *Bacteroidetes* is another important component of the gut microbiota that can degrade relevant carbohydrates from secretions of the gut, as well as high molecular weight substances (*Thoetkiattikul et al., 2013*). The proportion of *Bacteroidetes*, which was stable in most samples we collected except E5 (18.166%), would increase correspondingly with weight loss for mice or changes in fiber content in rural children's daily diet (*De Filippo et al., 2010*; *Ley et al., 2006*; *Turnbaugh et al., 2008*). However, the weight of the kestrel was increasing during the collection of E5 and E8. Additionally, although the kestrel underwent surgery on 4th July, the reason for the high proportion of *Bacteroidetes* in its fecal sample E5 were unclear. To characterize the basic composition and structure of the gut microbiota for the common kestrel more accurately, additional fresh fecal samples from healthy individuals should be collected in follow-up studies.

Furthermore, additional attention should be paid to the high ranking of *Patescibacteria* (0.543%) and *Deinococcus-Thermus* (0.504%) at the phylum level. *Patescibacteria* might be related to basic biosynthesis of amino acids, nucleotides and so on (*Lemos et al., 2019*). Members of *Deinococcus-Thermus* are known mainly for their capability to resist extreme

radiation, including ultraviolet radiation, as well as oxidizing agents (*Cox & Battista, 2005*; *Griffiths & Gupta, 2007*). The specific function of certain species in these phyla for the common kestrel should be studied by controlled experiments, detailed observations or more advanced approaches, as molecular biological techniques are developed.

In addition to the quantity of samples, living environment, age, sex and individual differentiation should also be considered as influencing factors, which would cause a degree of discrepancies at all levels in the gut microbiota. In addition, A comparison of wounded and healthy samples for the bacterial composition in the intestinal microbiota is another essential research direction that may provide additional information for wild animal rescue, such as important biomarkers that indirectly indicate potential diseases.

## CONCLUSION

In summary, using high-throughput sequencing technology in this study, we first characterized the elementary bacterial composition and structure of the gut microbiota for a wounded common kestrel in the BRRC, which could provide valuable basic data for future studies. Further research on *Enterococcus*, *Patescibacteria* and *Deinococcus-Thermus* should be conducted in the future with additional samples. The integration of other auxiliary techniques or disciplines, such as metagenomics and transcriptomics, could offer a deeper understanding of the function and mechanism of the gut microbiota, as well as the wildlife rescue.

## ACKNOWLEDGEMENTS

We sincerely thank the Beijing Raptor Rescue Center (BRRC) and Professor Limin Feng for sample collection of from the common kestrel.

### Funding

This work was supported by grants from the National Natural Science Foundation of China 31770410 and 31570381. The funders had no role in study design, data collection and analysis, decision to publish, or preparation of the manuscript.

### Grant Disclosures

The following grant information was disclosed by the authors:
National Natural Science Foundation of China: 31770410, 31570381.

### Competing Interests

The authors declare there are no competing interests.

### Author Contributions

- Yu Guan and Lei Bao conceived and designed the experiments, analyzed the data, prepared figures and/or tables, authored or reviewed drafts of the paper, and approved the final draft.
- Hongfang Wang performed the experiments, authored or reviewed drafts of the paper, and approved the final draft.
- Yinan Gong performed the experiments, analyzed the data, prepared figures and/or tables, and approved the final draft.
- Jianping Ge conceived and designed the experiments, authored or reviewed drafts of the paper, and approved the final draft.

## Animal Ethics

The following information was supplied relating to ethical approvals (i.e., approving body and any reference numbers):

The study was approved by the Animal Ethics Review Committee at Beijing Normal University (CLS-EAW-2019-026).

## DNA Deposition

The following information was supplied regarding the deposition of DNA sequences:

The data for our study are available in the NCBI Sequence Read Archive (SRA): PRJNA599118.

Data are also available at figshare: Guan, Yu (2020). ''Raw data''. figshare. Dataset https://doi.org/10.6084/m9.figshare.11691957.v1.

## Supplemental Information

Supplemental information for this article can be found online at http://dx.doi.org/10.7717/peerj.9970#supplemental-information.

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
