# Peer review of "The gut microbiota in the common kestrel (Falco tinnunculus): a report from the Beijing Raptor Rescue Center"

_PeerJ, doi:10.7717/peerj.9970_

## Round 0.1 · original submission · Major Revisions

Your paper has now been assessed by three expert reviewers, whose comments are appended below. You will see they all find merit in the study, but the reviewers and I all agree that the sample size of just a single kestrel is a major limitation of the study. I normally wouldn't consider a study on just a single individual, but I believe there is value here in there presence of longitudinal data.

In my opinion, you do not make the most of the data you have at hand, namely the ability to quantify temporal stability of the gut microbiome for the sequenced individual. Moreover, as reviewer 3 points out, this is not a healthy wild-caught kestrel but one that has transitioned from wild to captivity, with injury, and undergone various treatments. Therefore you have the ability to quantify the rate of change of microbiome in captivity from a wild baseline - which will likely be very useful to the wider field.

I would expect to see a graph of key microbiome traits, such as alpha diversity, over time - so the x axis being date-specific times that samples were taken. You may also want to do similar with metrics such as proportion of dominant taxonomic groups over time - for example to show 'succession' of microbiome through time. Please also indicate on the graph where interventions (e.g. surgery) took place, and what the interventions were. You can move rarefaction curves to supplementary material.

I look forward to seeing the revision.

Reviewer 1 ·

Basic reporting

Guan et al. conducted an interesting and important study of Common Kestrel gut microbiome. This study will be useful for the species conservation. I recommend this paper is accepted after the revision of my comments. Moreover, I strongly suggest the author should find a native English speaker and an expert in gut microbiota to improve the language.

Experimental design

Materials & Methods
The limitation is the samples from just 1 bird.
L 91. How the fecal could reflect the healthy condition? I am confused.
L134-136. PCA, PCOA and NMDS are redundant. You just need one approach.
L137-138. Reworte.

Validity of the findings

No comment.

Additional comments

General: The abstract should be rewrote.
L 35. Change composition and structure to community and delete microbiota.
L40. Move“A total of 28 phyla, 70 classes, 183 orders, 329 families and 681 genera were detected among the gastrointestinal bacterial communities.”to the first sentence of results.
L42. Remove this sentence “Further research....”.
L43. Remove this sentence “In addition....”
Introduction.
Reworte this sentence in lines 80-81.
Discussion
1.This section should be rewrote and language should be further improved.
2. The author mainly discussed the bacterial composition of each phylum and speculated the potential function. I think the author should also compare the results with the recently published paper of birds on mBio.
3. The author collected 9 samples from one animal in different days. And of course, the author observed the bacterial difference, so why this happen? I guess this may be linked to the healthy and treatment. This should be discussed.

Reviewer 2 ·

Basic reporting

This manuscript uses clear, unambiguous, English language, but uses certain terms quite casually without support. For example, in line 76 you reference the birds to have “special” attributes, but then do not follow up with what they are or how they are unique and thus having a different importance for microbiome studies.

Your introduction needs to start more broadly and the first paragraph should flow better instead of jumping from fact to fact when talking about kestrel natural history and biology. You should consider starting more broadly about the microbiome and then honing in on the species specific information. Expand upon the knowledge gap being filled.

In lines 72 through 76 you suggest that studies on these species have increased, however, after a preliminary literature review it appears that this is not the case. Please rephrase to state that in recent years, these species have been studied in regards to their microbiome, but that there isn’t a positive trend for each of these species.

For lines 72 through 76 consider adding Cockerham et al. 2019 and Fuirst et al. 2018 as an additional recent microbiome studies on Western and herring gulls. Especially for referencing microbial associations with urban and non-urban environments.

Please consider italicizing “Falconidae” in line 51

Where is your ICUN citation for lines 53-55

In line 80 do not use the word “elementary”

I would suggest replacing Figure 1 with a plot that shows the alpha diversity values for all metrics so that we can see the variation among samples. Rarefaction curves should supplemental material.

The text in Figure 4 appears stretched and blurry.

Experimental design

Molecular operational taxonomic units (OTUs) are often used for microbiome studies, but have been quickly shifting to using ASVs. OTUs use a 97% threshold, but you could avoid this and its ecological limitations, by using amplicon sequence variants (ASVs). Especially for a more descriptive microbiome study, using the most recent analytical methods is important. I would recommend following the protocol from Callahan et al. 2017. ASVs are a threshold-free metric and can be selected against databases using the DADA2 pipeline for example.

I do not see any mention of removing contaminants from samples or running negative controls. Was this done? If so, please state this explicitly.

In line 129 please explicitly state your rarefication depth.

Validity of the findings

The validity of these findings is quite limited due to small sample size. Please consider increasing the sample size for final publication.

The overall impact of this research is not mentioned in detail. The results are detailed and acceptable, but, there needs to be more emphasis on the significance of this study other than it being another avian microbiome study.

There should be more detail on where each of the nine birds were retrieved from and how that geographic and environmental origin could influence variation in the microbiome diversity and composition.

Analysis of this data is statistically sound, but I would highly encourage revising the analysis to use ASVs instead of OTUs. OTUs are not replicable between studies, and as a descriptive study, using ASVs would provide more detail and replicable methods for future research. See Callahan et al. 2017 as previously mentioned.

Additional comments

I commend the authors for their thorough research and expansive analyses from these samples. The manuscript is clearly outlined and written well. However, given that this is a more surface-level microbiome study, there should be much more emphasis on the importance for this. There is an overabundance of comparisons of taxonomy to other studies, but barely any link to why we need this information in the first place. Additionally, please consider revising the analysis to follow DADA2 ASVs rather than OTUs. Lastly, a descriptive microbiome study would definitely be strengthened by a larger sample size.

Reviewer 3 ·

Basic reporting

Although there are phrasing issues to improve clarity, the manuscript is clearly written, unambiguous, and professional English without grammatical errors is present throughout. The authors appear to have written the paper carefully.

The literature cited is robust and relevant. However, the introduction and background do not show the proper context, especially for a journal with a wide-ranging audience such as PeerJ. The introduction begins by introducing the study organism, the kestrel. However, the methods of the paper are much more far reaching- the kestrel here serves to show the importance of describing gut microbiota for general conservation and husbandry purposes, as well as to illustrate effects of surgery and drugs on an animal. The authors do nothing to place the unique value of their data in this context. This results in a paper that is less novel.

The structure of the paper is fine and follows PeerJ standard format. Figures are clear and concise, although I am not sure if the authors utilize colorblind friendly palettes.

Experimental design

No experiments are performed; the paper serves to report on gut microbiota in the kestrel. The research question is defined, and a knowledge gap, notably the lack of gut microbiota data in birds despite its potential importance, is defined. However, once again I think the authors miss the context of their own study- this isn’t just reporting on a bird that was captured and sampled. It is reporting on a bird that was injured, recovered, performed surgery on, and changed its regiment from life in the wild to captivity. But none of these aspects of the bird are highlighted with a context or relevant citations. I think that the unique background of the kestrel is what makes the study special- in addition to what the authors already provide- that birds are unique animals with an “extremely special body structure.” So while a knowledge gap is identified, I think another one is out there that is very beneficial to address.

The methods on the molecular and bioinformatic work are provided with sufficient detail to replicate. However, more detail on the sample collection would improve the manuscript. The kestrel’s unique history is not clear. The reader knows that the common kestrel was “carefully treated with several surgeries and drug therapies.” The details of these events could have a major impact on the microbiome, but we don’t know anything about what happened. For example, were antibiotics a part of the treatments?

We also need to be provided information about the samples. We don’t know anything about when they were collected relative to the life history of the kestrel. Information is vague- lines 90-92: “nine fecal samples that may reflect the real health condition were collected from the common kestrel after relevant treatments on different days.” This makes it sound to the reader as if all treatments were collected after different types of treatment, with the treatments taking place during different days. However, in the discussion, a few small details about sample collection are provided, lines 209-210 say “observed mainly in several samples that were collected during surgeries or drug treatments, such as E1 and E4.” This changes things, and implies that certain samples occur during treatments, but that others did not. If so, what was the duration of time between the last treatments? What treatments were received? Were any sampled before treatments? Any after treatment on the bird was complete?

Validity of the findings

A lot of data are provided in Tables, and data are available via FigShare. The speculation and claims in the discussion are appropriate. However, as with the introduction, I think the discussion should be framed within the context the authors provide- but also- with the link to learning about the microbiota of a bird with a unique life history. This will require the use of different citations to give context to this different point.

Additional comments

Introduction:

General comment: I think that the introduction’s format should be completely redone. Instead of starting general, and narrowing down to the study at hand, this introduction starts specific, and the context of the study is only given at the very end. The power of this study is that it characterized the gut microbiota as a way to potentially inform conservation efforts and husbandry of kestrels. It shows that gut microbiota analyses are a tool, that is often overlooked, during conservation. However, the introduction doesn’t get this point across- instead it starts with kestrels. The scope and impact of this study could reach far beyond kestrels to the general application of gut microbiota analyses coupled with conservation, however this more far-reaching point is lost in the introduction. I would flip, or reverse, the section.


Line 51: PeerJ is a journal with a general audience, may wanted to specify Aves or at least Falconiformes.

Line 54: Readers are likely not familiar with China’s scheme for classifying protected species. It is worth defining what second-class protected means. I do not have a context for this definition.

Line 55: I don’t think the kestrel is a typical opportunistic forager, I’m not sure what this means. Its hovering ability is exceptional.

Line 59: avoid the use of the term “wintering” as it is difficult to understand for readers based in more tropical latitudes or the southern hemisphere. “Non-breeding” is more appropriate. Also, the distinct predatory strategies at different times of the year here are mentioned, but not described. Would be good to describe the difference between the strategies.

Line 61: urban and rural kestrels have differences- but what are they? It’s better for the reader to know the characteristics of the differences (which has higher reproductive success, hunting success?) than to simply know that the differences exist in an undefined direction.

Line 65: probably don’t need this many citations. What does “comparatively comprehensive” mean? I don’t understand what was compared here. Are all these many studies comparing rural and urban?

Line 68: what is a research hotspot? May consider rephrasing.

Line 70: can be phrased better. “With the rapid progress of sequencing techniques, reports on gut microbiota have suggested important roles in…”

Line 73: probably don’t need to pluralize these species. Also, the species could be organized phylogenetically. Are there more raptors to add to this list (could remove non-raptors so that there are not too many citations)?

Line 76: “specialized body structure” could remove the phrase “extremely special.” Also, what about birds are extremely special? This is a general audience that is being addressed and may need more concrete detail.

Line 77: Additional used twice in this sentence. Overall this sentence’s phrasing could be revamped.

Methods:

Line 87-93: Definitely needs to be addressed: for sample collection, I think it is incredibly important to know what the kestrel’s diet was in captivity. Were any samples acquired before the kestrel was fed in captivity (and so would reflect the diet of a wild bird?). What subspecies of kestrel was used in this study?

Line 93: this is more of a descriptive study so I am not sure “experiments” is the correct word.

Line 130: I don’t think this is the correct way (“Team”) to cite R.

Results:

Line 141-147: Better to not have a single sentence paragraph, all one paragraph.

Line 150: Again, better to not have a single sentence paragraph. Can merge here.

Line 154: I don’t understand why this is remarkable.

Line 163: In which samples did these genera account for the largest proportion? I think it is sensible to compare these here.

Line 169-172: I think you mean “the hierarchical clustering trees showed”


Discussion:

General: I think you should start your discussion by interpreting the results, then zoom outwards to explain the context. Here, the discussion starts with the context. It would be better to save this until the end. So, I would invert the way that both the introduction and discussion are written.

Line 187-189: Very general sentences that don’t say much. Could be supported by citations.

Line 193: limited, or unknown? If limited, need a citation here.

Line 202: first mention of cockatiels, requires Latin binomial

Line 224: better phrasing “and an ancient symbiosis may be linked to the common ancestor of amniotes” or something of that nature

Line 234: I’m not sure what this sentence means- a hospital like environment contributed to the detection of this genus?

Line 237: The word “moreover” implies a connection to the next sentence. But here, you’re switching topics to a new genus.

Line 241: First mention of turkey, storm-petrel, requires binomial

Line 245: Do you mean the “presence of Bifidobacterium is closely related to”?

---

## Round 0.2 · Minor Revisions

Your manuscript has now been assessed by one of the original reviewers, who has identified numerous minor issues that require attention. In particular I agree with the reviewer that the findings lack the appropriate context at the start of the discussion. How do these data complement or advance existing data on the microbiome of wild birds and indeed other animals? Though you only have one animal in your study, and in a captive setting, longitudinal measures of microbiome structure are still quite rare in these fields

Reviewer 3 ·

Basic reporting

In my initial review, I discussed that the original manuscript contained clear, unambiguous, and professional English throughout. However, the new edited portions contain English that is written with many more issues than the original paper itself. In this way, the paper has actually regressed from its original submission. Basic mistakes (such as the use of "23th" for example) show that the edits likely did not receive the same amount of scrutiny as the original paper.

As I mentioned in my original review, the literature cited is robust.

Figures are shared properly. The paper is self-contained. It does not pose direct hypotheses as it is more of an exploratory study.

Experimental design

While the authors did incorporate my suggestion of mentioning the history of the kestrel, I do not believe that the changes are sufficient, and perhaps I was not clear about what I was asking for. The kestrel has a strange history, of being found injured and emaciated, as well as undergoing multiple treatments with different types of chemicals and surgeries. While the authors provide some anecdotally framed information about relevant samplings in the discussion, this is too challenging for the reader to piece together. I suggest mentioning these treatments clearly in the Methods, and mentioning all treatments corresponding to all sampling periods. This information would be greatly enhanced by a supplementary table detailing the sampling period, date, and relevant surgery, chemicals, or other notes (kestrel ate a mouse, for example) for that day. This would allow the reader to come to their own conclusions about the findings, rather than relying on the author’s selective interpretations in the discussion.

Additionally, the authors continue to use what I consider to be a comparatively weaker knowledge gap in the lack of knowledge about bird microbiota. I think the more interesting angle here is a medical one, regarding the kestrel's health and surgeries. However, if the editor and other peer reviewers do not find this to be a problem it could be published in the current context.

Validity of the findings

This is all fine.

Additional comments

General:

The authors may have misinterpreted my question about subspecies. Could the authors please identify the subspecies of kestrel they are observing? This is essential. I am not asking for a comparison of subspecies- just labellng the current kestrel that has been worked on.

Overall my main comment is that the care that went into the original paper is not reflected in the edits. The edited content is plagued by poor English and poor formatting (see literature cited).

Also very little context is given to the findings as a whole. The authors do a good job of explaining where specific findings fit into the field (this bacteria that we found also does this, and this, and so on) but not where their findings, in summation, fit into the field. Placing emphasis on this in the beginning of the discussion would be good. Again, I think that this study being performed on a captive bird that underwent surgery is a good place to start.

Line edits corresponding to the numbering system in the "track changes" version of the document.

Line 33:
Replace “extremely special” to “specialized”

Line 38:
Eliminate “in this study”

Line 41:
Eliminate “could also”

Line 48:
“Research” not “researches”

Line 48:
First sentence of new intro is a run on sentence, could break into two sentences

Line 53:
“Animals” plural. “has gradually become a” This sentence is not using proper English.

Line 70-72:
Re-read reviewer 2’s comment regarding this sentence. It still insinuates that studies on the species themselves have increased, not on birds as a whole. It still needs to be rephrased.

Line 72:
“With regard.”

Line 73:
Avoid casual abbreviations in scientific writing: “isn’t.” What does “positive trend” mean? Confusing.

Line 74:
You changed this sentence here, now change it in the Abstract.

Line 75:
This sentence is also not using proper English and needs to be edited. Also, not all birds are warmer than ambient temperature, for example in extremely hot desert environments.

Line 75-81:
I appreciate the additional information but this section is using poor English. I can’t put in the time to correct all of the English, the authors need to consult someone.

Line 86:
Should “BirdLife International” citation include a period in it?

Line 108:
Eliminate “the” in “the common raptors”

Line 109:
Eliminate “the” in “the food chains.” Eliminate the in “the common kestrels.” Is the kestrel a top predator? Presumably other larger raptors can prey on it.

Line 118:
“the injured common kestrel could not fly and was found…” Eliminate “first.” First is implied with “found”

Line 228:
Should “R Core Team” citation include a period in it?

Line 239-241:
Probably a mistake that could be eliminated.

Line 247:
“Are shown” as opposed to “were shown”

Line 282:
May want to relist these results with the smallest percentage first- it does the best job of illustrating your point, so may want to put it up front.

Line 304:
Eliminate “and depicted in”

Line 307:
“the variation” Eliminate “the”

Line 308:
Eliminate “in this case” - it is implied.

Line 312:
Eliminate “the”

Line 314:
“continues to be” Eliminate “was”

Line 318:
“were” past tense, because you have now characterized it.

Line 324:
Can probably just say “birds” here

Line 349:
“23rd” “June, the day after”

Line 350:
“22nd” June. This document needs to be reviewed by a native English speaker.
Line 351:
Unclear, revise English

Line 355:
“combined with”

Line 369:
Was “only” slightly lower. Otherwise using “although” in this sentence does not makes sense.

Line 374:
“Carbohydrates, sugar, and even by digesting fiber in some species”

Line 379:
Eliminate “special” ?

Line 382-390:
Again, information is appreciated but I don’t have time (or think it is my job) to make every English correction. Someone else needs to review the manuscript generally, but especially these new edited sections.

Line 393:
Don’t need to specify that Clostridium difficile belongs to Clostridium

Line 393-394:
Again, English issues.

Line 397:
“Remarkably.” Sentence is vague, need to back up with some numbers

Line 398:
Eliminate “the” in “the surgery” “While E5 was collected”

Line 414:
This is indicating relative abundance correct?

Line 419:
Do you mean “notably”?

Line 430-432:
Again, poor English. “Although the kestrel underwent surgery on 4th July, the reason for the high proportion…”

Literature cited:

There is inconsistent italicization and species capitalization of Falco tinnunculus in the literature cited.

I am not sure that the BirdLife International citation is formatted correctly, as I do not think a reader could retrieve information based on this citation.

The new citations are added very sloppily, without proper page numbers, often including URL links, and are inconsistent.

---

## Round 0.3 · Minor Revisions

Many thanks for making the requested changes to your manuscript. There are only a few minor issues that remain, and after these have been corrected I'd be happy to recommend your manuscript for publication

L42, 52, 120 & 240: I think the phrase 'animal protection' is far too vague. I think you mean that optimising the host gut microbiota could *possibly* be used as a conservation tool, but that's not clear here. More importantly your paper does not contain evidence that can be used to support this statement, so it certainly should appear in the abstract as if you have developed a 'novel strategy'.

L48: The opening sentence about advancement of sequencing technologies is not useful, in my opinion. Perhaps just open with the second sentence. edited to "Recent research on host-associated gut microbial communities have revealed their..."

L51: "processes"

L51: 'diseases' are neither critical nor basic 'processes'

L54: see above - remove vague statement about being a 'critical tool for animal protection'.

L69: "Although studies of the microbiome of these species have increased, there has been no corresponding trend in the study of each species." This sentence makes no sense.

L75" citations needed for these physiological generalisations

L119: As requested by a reviewer, please make this clear that this study is of a single kestrel sampled multiple times at a wildlife rescue centre. 'Wounded kestrel' isn't quite sufficient detail

L175: the citation you provide for R is for a book, not the software

L240: I think by 'protection' you mean 'conservation'

L389: "the" kestrel

---

## Round 0.4 · accepted · Accept

Thanks for making the required changes to the manuscript. I am now happy to recommend it for publication.